# Development of Questionnaire on the Sense of Workplace Involution for Newly Recruited Employees and Its Relationship with Turnover Intention

**DOI:** 10.3390/ijerph191811218

**Published:** 2022-09-07

**Authors:** Qi Chen, Yuzhu Zhang

**Affiliations:** 1School of Psychology, Inner Mongolia Normal University, Hohhot 011517, China; 2President’s Office, Hohhot Vocational College, Hohhot 010051, China

**Keywords:** sense of workplace involution, questionnaire development, reliability, validity, turnover intention

## Abstract

The term “involution” has become a buzzword in people’s daily communication and online discussion in recent years, and it has been used in many different contexts. However, the concept and structure of workplace involution are still unclear, and there is a lack of valid measurement tools. Objective: To explore the connotation and psychological structure of newly recruited employees’ sense of workplace involution, compile the Questionnaire on the Sense of Workplace Involution for Newly Recruited Employees, and examine its relationship with turnover intention. Methods: Through in-depth interviews with 30 newly recruited employees and based on qualitative research of the data collected by web crawler technology, the entries were compiled, 282 newly recruited employees were initially tested, and 571 newly recruited employees were given a formal questionnaire survey. The findings were analyzed using SPSS 22.0 and AMOS 24.0 for item analysis, exploratory factor analysis, and confirmatory factor analysis. Results: The newly recruited employees’ sense of workplace involution and its dimensions were significantly and positively correlated with turnover intention. Newly recruited employees’ sense of workplace involution was a significant and positive predictor of turnover intention. Conclusions: The reliability and validity of the Questionnaire on the Sense of Workplace Involution for Newly Recruited Employees met the needs of psychometric criteria, and the sense of workplace involution of newly recruited employees had a significant positive predictive effect on turnover intention.

## 1. Introduction

In 2020, the term “involution” was a search term popularized by a response to the time management strategy of a campus high school student, the most notable picture being a student riding a bike while operating a computer. The photo accidentally exploded through the network, and many Internet users call him the “king of involution”. Some people stated that this is “with life within the involution”, with the emphasis being on “involution”. Discussion surrounding this phrase suddenly caught wind, and it gradually became the mantra of contemporary young people. Young employees in various industries are always crying, “too much involution”, as if caught in a cycle of “stagnation despite their best efforts”. The term “involution” is a rather complex social problem that describes the social phenomenon of zero-sum competition in a limited development space [1]. It is noteworthy that when the concept of “involution” spreads to many industries, the social anxiety that it causes spreads uncontrollably [2]. However, contrary to the picture of a burgeoning discussion on the concept of involution, there is little clarity on what the term involution entails. The actual meaning of the term “involution” is rarely discussed, making the use of this concept rather arbitrary [3]. According to the Internet, almost anything can be termed “involution”. So far, the result is that the concept of “involution” has obvious generalization and entertainment tendencies. Despite the popularity of the term “involution”, we need to think about what we are talking about when we talk about “the sense of involution”. The specific connotation of “ involution” is constantly changing in the Internet world. What kind of life, work status, or group psychology of contemporary people does it reflect? How should we measure the sense of workplace involution of employees in the workplace today?

According to the verification of Wesson, the first person to propose the concept of involution was the German philosopher Kant, who proposed in his book *Critique of Judgment* that “involution” is the opposite of “evolution” and is a phenomenon of “regression”, describing the process of replication and complication upon the basis of a certain state [4]. Based on Kant, Alexander Goldenweiser described the cultural phenomenon of involution to be when society reaches a certain stage of development and reaches a certain final form that cannot be stabilized or transformed into a new form but merely continues its own refinement and internal complication [5]. Later, Geertz used the term “involution” to explain the economic form of Indonesia. Due to the limited amount of capital and land, agricultural development could not be expanded outward, and so a large number of laborers continued to enter the limited rice production industry, while the farming economy did not make any substantial breakthroughs, but only continuously increased in refinement [6]. In his study of rural economic development in northern China, Huang Zongzhi also used the term “involution” to describe the development of small farming economies in northern China, where the limited land and excessive labor force results in a form of “growth without development” [7]. From the perspective of organizational management, Lei Li and Wei Li outlined the phenomenon of “repetition without development” as an “involution” dilemma, which leads to the waste of material and mental resources and the deterioration of the management environment [8]. Since then, “involution” has become widespread and applied within Chinese academia, covering topics such as political science, economics, education, sociology, and many other disciplines, and it has been used to describe and explain other fields of related research [9]. Thus, it can be seen that involution is an open academic concept with a wide range of application areas that can be used to respond to, analyze, and explain many levels of social phenomena and social problems [10]. However, the definition of its concept is not sufficiently clear, and the academic community has still not reached any consensus on its exact meaning [11,12]. At present, there is a lack of theoretical research on the concept of “involution” in China, and there is a lack of relevant theoretical monographs. The concept of involution is not clearly defined and is used arbitrarily, resulting in a vague connotation of the concept and poor operability, which restricts its further development [3].

The original meaning of involution has been extended to more new connotations through its constant deconstruction and reconstruction within the current Internet context, which has been far removed from the original meaning of the word. It has expanded to include the workplace, which has not only triggered a strong resonance among young employees in the workplace but also reflects a certain aspect of general psychology, social mood, and spiritual expectations for numerous workplaces in today’s society [13]. In today’s competitive employment environment, the work motivation that is governed by the emotions of the involution will become more and more utilitarian, and competition between members of organizations will gradually evolve into work hours and work content that is based on the “amount” of competition. In the work process, the “superficial effort” of resource input and performance output is seriously duplicated, this has led to a lot of “ineffective competition” and “vicious competition”. In the competition that is spawned by the culture of involution, many employees who appear to be motivated on the surface are, in essence, not making progress in their work [14]. As some scholars argue, the phenomenon of superficial involution must be echoed by deeper psychology of involution [15], and the explosion of “involution” undoubtedly reflects a certain aspect of general psychology in today’s society. This study attempts to extend the discussion on the concept of involution to include individual psychology and behavior, and the “sense of involution” that describes the experiences and feelings that are brought to individuals at the psychological level during the objective process of involution, which is the repetition and continuation of negative experiences of young people over a period of time. The formation of the sense of involution stems from the accelerating changes of the current era, the hyper-intensified development of network technology, and the perceptual ecology of network communication, based on these, that intense stress, burnout, large-scale anxiety, and loss are easier to germinate [16]. The current state of involution in the workplace has made many young employees who have just started to feel a deep sense of involution in the workplace, which is highly consistent with the helpless and confused situation faced by most young people. The “sense of workplace involution” is an emotional resonance of employees in a situation of overwork and increased competition.

Recently, the topic of “996” has been featured on Weibo (a popular online social media in China), and it not only reflects the prevailing culture of overtime in the workplace, but it also hides the cruel truth of workplace involution. The concept of “the sense of workplace involution” is used to describe the current work content of workers, which means that the work content of workers does not have much meaning for the breakthrough of work goals and self-improvement, but workers are forced to participate because of increasingly strong competition. Such work content can only give people the lowest meaning for making a living, is helpless, and cannot give more creative value [17], which eventually puts individuals into a state where it is difficult to grow and break through themselves [18]. As the anthropologist Xiang Biao mentioned, for the current “working workers”, work seems to be a solution to material poverty, but it presents itself as a “poverty of meaning” [17]. Most people today are still entering the bottom of the meaning ladder, and they are making a living at the expense of their lives and health. If you don’t want to be eliminated, you have to work harder, but the effort and reward are not proportional, and ineffective efforts are always provoking people’s anxious nerves [19]. The sense of workplace involution is the experience of involution as a practitioner in a particular work environment that is wrapped up in involution, or the helpless feeling of having to engage in a low level of defensive complexity in order to survive in the workplace, with no way to exit, no way to disengage, and no way to benefit. This is like a hundred boats fighting for the flow of water, “if you don’t advance, you retreat, slow advance is also retreat”.

With the advent of the knowledge-based economy, the human resource team of enterprises is becoming increasingly highly educated, young, and diversified. Newly recruited employees generally have higher education levels and a strong desire to realize their values, but they are still at an early stage of career exploration, and compared with most veteran employees, they are not clear about their future development directions and have vague self-orientation and career goals [20]. However, they are eager and willing to try new experiences and challenges brought about by their work, and in an increasingly competitive workplace environment, they are not willing to lag behind other colleagues, so they often have a “continuous striving drive”. This “drive”, which is fueled by the Internet media, maintains people’s experience of anxiety [21]. In addition, these young employees, who have just entered the early stage of their careers, often have great pressure to survive, and may even feel the need to urgently adapt to the workplace, by means of “vicious competition”. This often results in more anxiety and fatigue and places them at risk of irrational thinking and voluntary or forcible workplace participation. They become deeply troubled by the sense of workplace involution. At present, researchers are defining this group of newly recruited employees in terms of their working hours, but domestic scholars have not been able to unify the scope of time. Newly recruited employees can be divided into the following two categories. Newly recruited employees are defined as those that have joined the enterprise for less than one year. Second, employees that have remained with the enterprise for less than three years can be regarded as newly recruited employees [22]. Combining the views of many scholars, and in order to broaden the object of research, this study defines the working time of newly recruited employees as those that have entered the company within three years or those who are in the early stages of career exploration.

Amid the raging COVID-19 epidemic, the chaotic international environment, and the changing economic order, it seems that all industries are carrying a heavy load [23]. In some high-intensity industries, the employees of enterprises must sacrifice their rest times and work long hours in a high-intensity environment in order to improve their material living standards and work performance. Overtime is closer to an invisible state, without explicitly being defined as overtime, but developing into overtime culture and work. The environment forces the individual to continue working. Young people who are burdened with life, they can only join this “involution” [24], which can also lead to a series of problems such as leaving work, work–family conflicts, and reduced job satisfaction [25]. Such a sense of workplace involution affects physical and mental health, as well as the subjective well-being of young employees, making their energy levels and body suffer, and thus raising the idea of leaving their current positions. Mobley views separation as the process by which an individual terminates his or her interest in the organization and membership in the business [26]. Although the turnover intention is not necessarily the same as an employee’s departure behavior [27], as the final step in the exact departure process [24], it is a strong predictor of an employee’s real departure behavior. Research on turnover intention can provide reasonable suggestions for the development of enterprises, which is important for enterprises to reduce their turnover rates and prevent brain drain. Accordingly, we should explore the reasons for the proliferation of the sense of workplace involution, explore the relationship between the sense of workplace involution and turnover intention, reduce the frequency of the latter, and understand the organizational behaviors that are related to the sense of workplace involution, especially about waste of resources and unhealthy competition. This helps to implement effective countermeasures to help newly recruited employees cope with the constant stream of sense of workplace involution, prevent employees’ turnover intention induced by the explosion of the sense of workplace involution, maintain employees’ rationality and enthusiasm for their work, and retain talent for the company [28]. Therefore, this study proposes the research hypothesis that newly recruited employees’ sense of workplace involution has a significant positive predictive effect on turnover intention.

At present, academic discussions and considerations on the sense of workplace involution of newly recruited employees in the new era are far from adequate, especially in psychological research. Compared with the research on the theory and reviews of involution, in the field of psychology, there is a paucity of research on the psychological aspects of involution of individuals and groups in the workplace, and the empirical research related to this is relatively weak, and there is a lack of instruments with good reliability and validity to measure employees’ sense of workplace involution. Hence, this study takes newly recruited employees’ work as being the main research object and uses in-depth interviews and web crawlers to collect responses on contemporary newly recruited employees’ understanding and definition of the sense of workplace involution. The connotation and structure of workplace involution were analyzed, and based on the structure, compile the Questionnaire on the Sense of Workplace Involution for Newly Recruited Employees. This provides a validity tool for assessing the current sense of workplace involution of newly recruited employees. Moreover, using this questionnaire, we explored the influence of workplace involution on turnover intention and understood the overall characteristics of workplace involution and turnover intention at this stage. It is also important for understanding the work status of newly recruited employees, and retaining talents for the company, improving their sense of job satisfaction and well-being, reducing the turnover rate, and stabilizing the human resources of enterprises.

## 2. Research Methods and Procedures

### 2.1. Research Subjects

This study followed the “purposive sampling” principle of qualitative research [29]. Intensive sampling was employed to find users who commented frequently on topics related to “involution” from their surroundings and from the Internet; i.e., those who were very intensive and rich sources of information [30]. According to Lincoln and Guba [31], the sample size for interviewing purposes should be larger than 12. Therefore, the total number of pre-interview subjects for this study was 3. In the formal study, a total of 30 respondents were recruited, 16 females and 14 males, ranging in age from 25–29 years old, with education levels ranging from a specialist and below to Ph.D., from different provinces in China. Their occupations ranged from being general employees in various enterprises and institutions to Internet workers, engineers, etc. Their years of work experience were 1–3 years. Respondents who participated in the study were paid 30 RMB after the interview.

The questionnaires were sent and collected through the online method by combining purposeful sampling and convenient sampling methods. Most of the test subjects were Newly Recruited Employees from the Inner Mongolia region. A total of 546 questionnaires were collected, of which 282 were valid, and the effective recovery rate was 51.65% (Sample 1).

The selection of formal measurement subjects was extended to all over the country, and questionnaires were sent and collected through online completion using purposeful and convenient sampling methods. A total of 632 questionnaires were returned, of which 571 were valid, and the valid return rate of the questionnaires was 90.35% (Sample 2).

### 2.2. Research Methods

#### 2.2.1. Questionnaire on the Sense of Workplace Involution for Newly Recruited Employees

Questionnaire on the Sense of Workplace Involution for Newly Recruited Employees was used to investigate the current situation of sense of workplace involution among newly recruited employees, and qualitative research and questionnaire methods were used to collect information and data. The questionnaire was developed as follows: firstly, the interview begins by informing the interviewee of the content, purpose, and procedures to ensure informed consent of the respondents and then conduct the interviews. Before the formal interview, conduct pre-interviews with three interviewees to ensure that the interview questions are clearly stated, unambiguous, and can stimulate the interviewees to express themselves. Then, semi-structured interviews were conducted with 30 newly recruited employees, with the main questions being, “What do you consider to be the “sense of workplace involution?” Have you ever experienced the sense of workplace involution at work?”; “What do you think are the factors that contribute to the prevalence of the sense of involution in the workplace?”; and “How do you think we should deal with the sense of workplace involution that happens from time to time?”. With the consent of the interviewees, the entire interview was recorded using a tape recorder and then transcribed into text and proofread. To obtain a large amount of research-related and even critical background information from the content of the interviews and to understand the respondents’ knowledge of the research topic and related topics, each interview lasted approximately 40–60 min. In this study, the shortest interview time for a single respondent was 25 min and the longest was 83 min, for a total interview time of 1300 min. Meanwhile, using “Descendant Collector”, a data collection software, to search in Sina Weibo, Zhihu.com and Bilibili, and other network platforms, such as “social involution”, “the sense of involution” and “the sense of workplace involution” and other keywords, the data collection template contains a number of data results such as keywords, title, content, release time, author, etc. The format of exported data is uniformly stored in Excel. Big data samples extracted from the Internet are mostly sourced from “local concepts”, which are compatible with qualitative research [30]. The relevant web texts collected by this data collection software were categorized and organized, and analyzed and organized using the open-ended coding, associative coding, and core coding methods of grounded theory. Grounded theory is a research method that abstracts and generalizes new concepts and ideas from human empirical facts through a systematic research process. Three-level coding is the key to analyzing data. The first-level coding is open coding, which encodes the content related to the connotation of workplace involution in the original material sentence by sentence. The secondary coding is the associative coding, which understands the internal connection between the open coding and merges the coding with similar meaning. The third-level coding is the core coding, which repeats the text data Reading and thinking, combined with associative coding to form core categories [30]. Based on the above, the connotation and structure of workplace involution were obtained, and the initial version of the Questionnaire on the Sense of Workplace Involution for Newly Recruited Employees was compiled using this basis. Finally, a subset of subjects (Sample 1) was selected to pre-test the questionnaire for item analysis and exploratory factor analysis, and the questions were further screened. A new group of subjects (Sample 2) was then recruited, and the new subjects were formally measured using a formal questionnaire with screened questions, and confirmatory factor analysis was used to check the structural validity of the questionnaire.

The final official questionnaire, with a total of 16 questions (Appendix A), included 4 dimensions: a sense of scarcity of resources, a sense of compelled commitment, a sense of the futility of effort, and a sense of negative experience. The questionnaire was scored on a 5-point scale, (1 = “strongly disagree”, 2 = “disagree”, 3 = “neutral”, 4 = “agree”, and 5 = “strongly agree”); the higher the score, the higher the degree of workplace involution feeling of the employees. The questionnaire had good reliability and validity, with Cronbach’s α coefficient of 0.963 for the total questionnaire, and 0.876, 0.871, 0.867, and 0.870 for the sub-questionnaires, respectively.

#### 2.2.2. Turnover Intention Scale

The Tendency to Leave Scale (TIS), developed by Mobley, was used. It consists of four statements, such as, “I may quit my current job and join a new organization”. The questionnaire was scored on a 5-point scale, (1 = “strongly disagree”, 2 = “disagree”, 3 = “neutral”, 4 = “agree”, and 5 = “strongly agree”). The higher the score, the stronger the employee’s turnover intention. Chinese scholars translated the turnover intention scale compiled by Mobely for research, and its Cronbach’s α coefficient was 0.930 [32]. The Cronbach’s α coefficient for this scale in this study was 0.856.

### 2.3. Research Procedures

This study combines qualitative and quantitative research and adopts a hybrid research approach to improve research efficiency. The combination of the two can make the ecological validity of psychological research effective, and enable psychology to play a real practical role in solving actual social problems. First, qualitative interviews were conducted with newly recruited employees, and the contents collected by the web crawler were organized together as textual information for the qualitative study. The questionnaire questions were collected through qualitative analysis of the text and the initial questionnaire was developed. In the second step, a pilot test is conducted, and the questionnaire is revised as necessary. In the third step, a formal measurement is conducted to validate the structure of the questionnaire and to check the reliability and validity of the questionnaire and whether it meets the requirements of psychometrics. The qualitative interviews and the web-based text analysis were conducted from May to August 2021, the pilot test was conducted in October 2021, and the formal measurement was conducted in November 2021.

### 2.4. Statistical Analysis

NVivo12 software (QSR International Pty Ltd., Melbourne, Australia) was used to classify and code the textual information, and SPSS 22.0 (SPSS Inc., Chicago, IL, USA) and AMOS 24.0 (SPSS Inc., Chicago, IL, USA) statistical software were used to process the data.

## 3. Research Results

### 3.1. Results of Qualitative Analysis

#### 3.1.1. Coding Results

The specific coding process of the connotation of newly recruited employees’ sense of workplace involution is shown in Table 1. Through the analysis of online texts and qualitative interviews, 352 semantically clear meaning units were obtained by open coding, and 9 conceptual words were obtained by associative coding, and these were further categorized into 4 core categories after core coding: a sense of resource scarcity, sense of compelled commitment, sense of the futility of effort, and sense of negative experience. The content and structure of newly recruited employees’ sense of workplace involution were well-summarized into the four core categories, and the preliminary concept of newly recruited employees’ sense of workplace involution was formed based on this study.

#### 3.1.2. Confidence of Coding

In this study, to ensure the reliability of the coding, another master’s degree in psychology who had done a qualitative study was invited to discuss and select the same information for coding and examine the coding agreement between the researcher and the rater: rater agreement = number of mutually agreeing codes/(number of mutually agreeing codes + number of mutually disagreeing codes). The percentage of agreement in coding was calculated to be 79%.

### 3.2. Test Results

#### 3.2.1. Distinguishability of Questions and Discrimination Index

The correlation coefficient (product-difference correlation) between the score of each question and the total score of the scale was used as the index of discrimination. Forty-four out of the 47 questions had a correlation coefficient of 0.4 or more, and three were less than 0.4. Three questions with correlation coefficients of less than 0.4 compared to the total score of the questionnaire were deleted, namely Questions 21, 28, and 32. The total score of the questionnaire was calculated using the decisive value method, and it was then ranked from highest to lowest, with those scoring in the top 27% being in the higher group and those scoring in the bottom 27% being in the lower group. An independent sample *t*-test was conducted based on the differences between the high and low groups of each question, and all questions were significant (*p* < 0.01), indicating that all the discriminant indices met the requirements, and so all of them were retained.

#### 3.2.2. Exploratory Factor Analysis

In this study, we chose principal component analysis and maximum variance with orthogonal rotation to conduct exploratory factor analysis on 44 entries, taking a principal component analysis to extract the common factors and using a maximum orthogonal rotation of variance, with a KMO value of 0.949 and Bartlett’s spherical test chi-square value of 7602.228 (*df* = 946), *p* < 0.001, which indicates the existence of common factors among the topics. This was suitable for exploratory factor analysis. After the first pivot, nine factors with feature root values greater than one were obtained, and the cumulative variance explained was 63.717%. The criteria of “the loading of a topic on a common factor are greater than 0.4 and the loading on other common factors are not greater than 0.4” were used to sequentially delete the topics that did not meet the requirements, and exploratory factor analysis was performed again after each topic was deleted. Finally, a total of 28 topics were deleted and 16 topics were retained, with 4 common factors with eigenvalues that were greater than 1. The variance of the interpretation rates were 17.897%, 17.045%, 16.454%, and 12.267%, and the cumulative interpretation rate was 63.663%. The factors were named according to the significance and conceptual dimensions of the topics with higher loads (see Table 2). The results showed that there were 16 entries in the sense of workplace involution questionnaire for newly recruited employees, comprising 4 entries for the sense of scarcity of resources, 4 entries for the sense of compelled commitment, 4 entries for the sense of the futility of effort, and 4 entries for the sense of negative experience.

#### 3.2.3. Factor Naming

The four factors obtained from the exploratory factor analysis corresponded to the four core categories of the qualitative analysis and the four factors obtained from the exploratory factor analysis were named according to the factor attribution of each topic and the results of the core coding of the qualitative analysis. The four topics included in Factor 1 reflected that the scarcity of quality resources in the workplace brought about great competitiveness and survival pressure to the employees, and so they were termed “sense of scarcity of resources”; The four items included in Factor 2 reflected the nature of the workplace involution environment, where a lot of investment is not voluntary, with many work tasks being helpless choices within the involution environment, and so it was termed “sense of compelled commitment”. The four topics included in Factor 3 reflected that there is no substantial gain from the forced workload, and so it was named “sense of the futility of effort”. The four topics of Factor 4 reflected the negative feelings brought about by the individual’s experience in the workplace, including confusion and helplessness, and heartache and pain, and so it was termed “sense of negative experience”.

#### 3.2.4. Confirmatory Factor Analysis

According to the results of the exploratory factor analysis, the formal questionnaire was set as a four-factor model containing 16 observed variables, and AMOS 24.0 software was used for confirmatory factor analysis, using maximum likelihood estimation, and the standardized parameter coefficients were between 0.70 and 0.85, which met the requirements. The results of the confirmatory analysis determined that the indicators of the newly recruited employees’ sense of workplace involution were better. See Figure 1 and Table 3 for details.

### 3.3. Reliability and Validity Tests of the Questionnaire

#### 3.3.1. Reliability of the Questionnaire

The internal consistency coefficient (Cronbach’s α), split-half reliability, and test-retest reliability of the overall questionnaire and each sub-questionnaire were examined. The results showed that the α coefficient of the overall questionnaire was 0.963, and Cronbach’s α coefficients of each sub-questionnaire were 0.876, 0.871, 0.866, and 0.870 respectively. The split-half reliability of the total questionnaire was 0.961, and the split-half reliability of each sub-questionnaire ranged from 0.863 to 0.879. The test-retest reliability of the 40-day interval was 0.948, and the test-retest reliability of each sub-questionnaire ranged from 0.830 to 0.859, with all values being greater than 0.80, indicating that the questionnaire has good reliability indicators. See Table 4 for details.

#### 3.3.2. Content Validity of the Questionnaire

The questions in this study were prepared in the content of the qualitative analysis, and the clarity and generality of the questions were checked by experts and multiple people. For example, “Often blindly imitate colleagues in their work, because only when everyone is at the same level will they not be eliminated”. Revise to “Colleagues often blindly imitate each other at work, because only when we are all at the same level will we not be eliminated”. A certain range of preliminary tests was conducted at the beginning of the formal distribution of the questionnaire, and the subjects were asked whether the linguistic descriptions were understandable and whether there were ambiguous contents after they answered the questionnaire, so that the ambiguous questions were revised and deleted, which could ensure that all questions could accurately. This ensures that all questions can accurately express the meaning of the questions, thus ensuring that the questionnaire has a certain degree of content validity.

#### 3.3.3. Structural Validity of the Questionnaire

According to the exploratory factor analysis, the structural dimensions of the sense of workplace involution of newly recruited employees were basically consistent with theoretical conceptions, indicating that this questionnaire has good structural validity. In addition, the correlation between the sub-questionnaires and the correlation between the sub-questionnaires and the total questionnaire was used to test for the structural validity of the questionnaire. The correlation between the total questionnaire and the sub-questionnaires was positive (*p* < 0.001) and exceeded the correlation level between the sub-questionnaires, indicating that the structural validity of this questionnaire meets the requirements of psychometric criteria. For details, see Table 5.

### 3.4. The Relationship between Newly Recruited Employees’ Sense of Workplace Involution and Turnover Intention

#### 3.4.1. Descriptive Statistics of Variables

The demographic variables for this study are in Table 6. *t*-tests and one-way ANOVA were conducted for gender, education level, nature of work unit, and region of workplace affiliation in each dimension of sense of workplace involution, and the results are shown in Table 6. Considering that there are fewer people and more detailed categories in other categories, the comparison differences are not meaningful, and they will not be analyzed here. Descriptive statistics for each variable are shown in Table 7.

#### 3.4.2. Correlation Analysis between the Sense of Workplace Involution and the Turnover Intention among Newly Recruited Employees

Pearson product difference correlation was used to examine the correlation between newly recruited employees’ sense of workplace involution and its dimensions and turnover intention scores, and the results showed that there was a significant positive correlation between turnover intention and newly recruited employees’ total score of the sense of workplace involution and each dimension. For details, see Table 8.

#### 3.4.3. Regression Analysis of Factors Influencing Turnover Intention

In order to further investigate the influencing factors of turnover intention, a hierarchical regression analysis was carried out with turnover intention as the dependent variable, and demographic variables and the sense of workplace involution as independent variables. First, gender, educational level, nature of work unit, and the area of workplace constitute the first stratum and enter the regression equation; secondly, the sense of workplace involution enters the regression equation as the second stratum. As shown in Table 9, newly recruited employees’ sense of workplace involution has a significant positive predictive effect on turnover intention.

## 4. Discussion

This study focuses on the sense of workplace involution that is prevalent in the current social work environment, and it adopts a bottom-up strategy to develop the Questionnaire on the Sense of Workplace Involution for Newly Recruited Employees, based on the results of qualitative analysis, using the meaning units obtained from open coding as the main source. After exploratory factor analysis and confirmatory factor analysis, it was finally found that newly recruited employees’ sense of workplace involution was mainly reflected in four aspects: a sense of scarcity of resources, a sense of compelled commitment, a sense of the futility of effort, and a sense of negative experience. This four-factor model fits well with the findings of qualitative research and has high reliability and validity, which is in line with the requirements of psychometrics and has both a theoretical basis and a realistic foundation.

The descriptive statistical analysis of the sample showed that the scores of the sense of workplace involution and the turnover intention of newly recruited employees are moderately high, among which the score of the negative experience dimension is the highest, which indicates that the sense of workplace involution brings more harm to individuals. Gender does not affect the sense of workplace involution and the turnover intention of newly recruited employees. The possible reasons for this result are that today’s workplace is full of stress, and both men and women face competition for limited resources in the workplace, many times income is not proportional to effort, job burnout, and anxiety. From the analysis of the data concerning education level, it was found that with an increase in education level, the sense of workplace involution and the turnover intention the job decreased for the newly recruited employees, which indicates that we can reduce negative experiences in the workplace by improving education levels and increasing our own competitiveness and irreplaceability. From the perspective of the nature of the work unit, newly recruited employees in private enterprises, foreign-funded enterprises, and Sino-foreign joint ventures have a higher sense of workplace involution and turnover intention. This is because of the high work pressure in these types of enterprises, the more complex relationships between colleagues, and the fierce competition between them, which creates more tendency for employees in these enterprises to leave compared to employees in state-owned enterprises [33]. Newly recruited employees’ sense of workplace involution and their turnover intention are low within state-owned enterprises, public institutions, state administrative organs, and the government because the working environment in the system is relatively stable. The sense of workplace involution and the turnover intention of newly recruited employees are also affected by their area of the workplace, among which, the sense of workplace involution and the turnover intention of newly recruited employees in eastern areas are greater than those in the northeast, central, and western areas. Because the eastern area consists of the developed coastal economic area, there are many economically developed cities and more development opportunities, and so it attracts more young people, and when the competition pressure is greater, the sense of workplace involution is stronger.

The term “involution” is not only a concise summary of the current mode of social development, but it is also a sharp description of the social competition that exists in society at large. The corresponding dilemma is the contradiction between the needs of individuals and their limited social resources, as numerous individuals expect to compete for more resources. In this trend, “involution” competition also refers to irrational competition, which puts people in an awkward “dilemma” as it becomes a more normalized way of life. Among them, the dimension of the sense of scarcity of resources emphasizes that the need for newly recruited employees to obtain quality resources in the workplace will always exist, and with the development of society and the continuous improvement of people’s knowledge, this demand will become greater, and the competition will become increasingly intense [1]. At the same time, “involution” reflects the competitive nature of the personal development process in young people. An increase in competitive pressures is the root cause of the social environment of the sense of workplace involution [34]. Since there are only a few resources or far more competitors than the resources can carry under the same clear goal [35], people are forced to obtain resources by “involution” under the great pressure of survival. The dimension of the sense of compelled commitment refers to a choice that newly recruited employees are involuntarily compelled to make under workplace competition, and participants who are involuted rarely quit the process of involution easily, because, for most young employees in the workplace, involution is a road of no return [36]. In comparison to their peers, people must contribute more, and the rewards do not match the contributions. The dimension of the sense of the futility of effort emphasizes that inefficient work not only saps employees’ will and enthusiasm, but also damages the company’s resources and interests [37]. In a workplace environment full of involution, it is futile to obtain enough opportunities and resources, even if one pays more, due to the constantly rising threshold; This is the dilemma of involution, where workplace employees repeatedly work hard without creating new value, and they cannot obtain real growth within such a dilemma. The dimension of the sense of negative experience refers to the negative emotional experience of being “involution”, people who experience a sense of involution often feel a lack of spirit, precisely because he or she is in an unstable competitive state for a long period of time [18]. This is a cruel experience of helplessness, irritation, heartache, confusion, and the inability to quit, even though it is painful. In today’s society, involution is a product of the moment, and it will continue to persist in the future. The real danger of involution is that what is done gradually loses its value, leading to a decline in our level of happiness and achievement. The sense of workplace involution is both a drain on personal energy and a serious detriment to productivity.

No matter the field, excessive internal consumption, and vicious competition are a waste of resources. However, it is also very inappropriate to call all of the efforts, struggles, and competitive behaviors “involution”. When we cannot define the meaning of involution, we cannot use it correctly. In fact, the so-called “involution” of some Internet users is more of a certain way of adrift complaining, a negative response to their own situation, which does not mean that they have actual insight into this concept as a social phenomenon [38]. We should be vigilant in dealing with workplace environments that are characterized by “involution”, which is a stage of depression and atrophy in the individual life development of contemporary youths and may cause stagnation and atrophy in the development of societies and countries when the sense of workplace involution spreads among young employees. We should take this opportunity to work harder, maintain a positive attitude and rational thinking, and show the “power of upward mobility” in young people. We should seek new tracks of personal interest, break single-minded orientations so that the social value of diversity is kept, and adjust our mindset when facing pressures. Additionally, one person’s life does not have to be the same as another person’s, and status, wealth, and power are not the only criteria for success in life.

In summary, the scale comprehensively measures the sense of workplace involution that is experienced by newly recruited employees in the workplace using four dimensions, and the scale as a whole is concise and condensed. This is not only supported by empirical data, but it also reflects the difficult situation and the strong emotional resonance of newly recruited employees in the workplace. These four different aspects of involution are what we need to actively work to change during this time. Young people in the workplace are increasingly questioning the meaning of work, and they find it difficult to obtain a sense of satisfaction and accomplishment at work, which has led to their growing dislike of the “struggle culture” and the reality of the “overtime culture” in the workplace. This workplace state causes the daily life of young people to be occupied by work, and in the fast-paced work atmosphere, it not only reduces the efficiency of employees, but also exhausts their enthusiasm for work, and they gradually lose the joy of work. More and more young employees can feel the growing sense of workplace involution. This anxiety and confusion are difficult to deal with in the short term, which also leads to an increase in young people’s turnover intention and even turnover, and even the behavior of leaving. These factors are not conducive to long-term personal growth and development, both for the enterprise and for its employees. The turnover intention has a good predictive effect on employees’ departure behavior, and the occurrence of departure behavior often brings additional costs and adverse effects to the organization. Consequently, it is important to clarify the potential factors that cause employees’ turnover intention. This study investigated whether there is a correlation between the dimensions of the sense of workplace involution in newly recruited employees and the turnover intention, based on the development of a questionnaire, to understand the process by which the sense of workplace involution of newly recruited employees influences the turnover intention. The results showed that the sense of workplace involution and its intrinsic dimensions were significantly and positively correlated with the turnover intention, and the sense of workplace involution of newly recruited employees had a significant positive predictive effect on the turnover intention. In other words, the higher the level of the sense of workplace involution of newly recruited employees, the higher their turnover intention. This suggests that we should pay attention to the psychological health of young employees, reduce the sense of workplace involution of young employees, actively improve the current situation of the competition among employees and create a rational competitive atmosphere; introduce diversified talents, stimulate employees’ enthusiasm for innovation and create challenging experiences for employees; guide newly recruited employees to improve their career planning, and enhance their sense of career goals in order to achieve continuous development.

The innovation of this study is that from the theoretical research aspect, it is a complement and extension of the theories in the field that are related to involution. Most of the previous studies on involution exist in the field of sociology, and fewer studies have explored it from the psychological perspective of individuals. This paper is a study of newly recruited employees, who are an important force in the company, focuses on the mental health of young employees, deconstructs the concept of the sense of workplace involution, proposes a clear connotation and structure, and extends the study of the sense of workplace involution in the field of management psychology. Second, in terms of research technology, in the context of the big data era, the use of web crawler technology to obtain a large amount of relevant content will make the research data more extensive and comprehensive. In this study, the third-party collection software “Descendant Collector” was used to efficiently capture and organize a large amount of text data. Thirdly, from the perspective of practical operations, this study provides some new thoughts and findings for the human resource management of enterprises. In a time of increased competition, it is crucial to improve productivity, reduce waste, and retain talent, and business managers should pay attention to this phenomenon and provide the right guidance in terms of policies and mechanisms.

The limitations of this study are that, firstly, the target population of this study is newly recruited employees, i.e., young employees who have been employed for less than three years. However, the definition of “newly recruited employees” varies from industry to industry; consequently, in future studies, the target population should be limited to a specific field to explore the sense of workplace involution in that field in a more focused manner. Secondly, due to the different cultural backgrounds of each country, this study lacks a comparative study of the differences between the collectivist culture of East Asia and the individualist culture of Europe and the United States, and future studies can further explore the cultural differences of “the sense of workplace involution”. Third, this study collected data based on interviews and Internet crawler technology for the construction of rooting theory, but due to limited human and time resources, the textual data and data information obtained are still relatively limited, and future research can further increase the collection of sample data to enhance the reliability of the research results and further improve the scale. Fourth, because there are fewer instruments measuring the sense of workplace involution of newly recruited employees, the test of calibration validity was not conducted in this study, but in terms of structural validity, this study ensures the scientific validity of structural validity through theoretical and confirmatory factor analysis.

## 5. Conclusions

This study found that the sense of workplace involution of newly recruited employees is a structure that includes four factors: the sense of resource scarcity, sense of compelled commitment, sense of the futility of effort, and sense of negative experience. The reliability and validity of the Questionnaire on the Sense of Workplace Involution for Newly Recruited Employees met the needs of psychometric criteria, and it can be used as a tool to measure the sense of workplace involution of newly recruited employees. The sense of workplace involution of newly recruited employees had a significant positive predictive effect on turnover intention.

## Figures and Tables

**Figure 1 ijerph-19-11218-f001:**
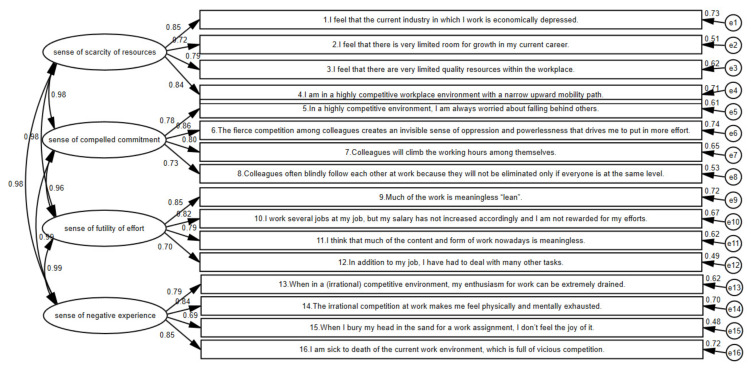
Confirmatory factor analysis of sense of workplace involution.

**Table 1 ijerph-19-11218-t001:** Coding of the connotation of the sense of workplace involution for newly recruited employees.

Open Coding	Frequency	Associative Coding	Frequency	Core Coding
Large population and many competitors	25	Scarcity of quality resources	128	Sense of scarcity of resources
Limited career advancement and quality jobs	17			
Fierce competition and difficult to promote positions	14			
The majority of employees have a single career goal	24	Convergence of competitive goals		
The majority of employees have a single work goal	48			
Often can’t get off work on time	20	Paying more time	75	Sense of compelled commitment
Simple problems are infinitely complicated	17			
Working several jobs at the same time	7	Paying more energy		
Continuously improving academic qualifications	10			
Staying up late and working overtime and being Physically and mentally exhausted	13	Paying more physical and mental health		
Overtime takes up a lot of time for rest, recreation and social interaction	8			
Meaningless imitation and overtime	7	Learned but no substantial growth	45	Sense of futility of effort
Meaningless meetings are held frequently	6			
No breakthroughs and growth despite commitment	9			
Endlessly digging into research on the same problem	6	No or very little gain		
Income is not proportional to effort	17			
Social comparison causes anxiety	44	Decreased sense of happiness	104	Sense of negative experience
The helplessness of not being able to leave and not having anyone to benefit from	17			
Lack of job satisfaction and job well-being	10			
Decreased levels of both physical and mental health	7			
No enthusiasm for work, dislike the work environment	14			
Unable to achieve the realization of the value of life	7	Loss of meaning in life		
Lack of sense of meaning in life and work	5			

**Table 2 ijerph-19-11218-t002:** Results of exploratory factor analysis.

Sense of Scarcity of Resources		Sense of Compelled Commitment		Sense of Futility of Effort		Sense of Negative Experience	
Title	Payloads	Title	Payloads	Title	Payloads	Title	Payloads
24	0.736	40	0.738	33	0.745	60	0.809
23	0.697	25	0.712	41	0.712	61	0.738
20	0.610	30	0.627	29	0.627	59	0.735
19	0.421	39	0.544	26	0.544	65	0.619

**Table 3 ijerph-19-11218-t003:** Fit indices for the confirmatory factor analysis of the model.

	χ^2^	*df*	χ^2^/*df*	GFI	NFI	RFI	IFI	TLI	CFI	RMR	RMSEA
models	142.210	98	1.451	0.970	0.981	0.976	0.994	0.993	0.994	0.023	0.028

**Table 4 ijerph-19-11218-t004:** Reliability analysis of the sense of workplace involution questionnaire.

Dimension	Number of Projects	Internal Consistency Reliability	Split-Half Confidence	Retest Reliability
		(*n* = 571)	(*n* = 571)	(*n* = 96)
Sense of scarcity of resources	4	0.876	0.879	0.859
Sense of compelled commitment	4	0.871	0.871	0.849
Sense of futility of effort	4	0.866	0.863	0.843
Sense of negative experience	4	0.870	0.865	0.830
Sense of workplace involution	16	0.963	0.961	0.948

**Table 5 ijerph-19-11218-t005:** Construct validity of the sense of workplace involution questionnaire for newly recruited employees.

	Sense of Workplace Involution	Sense of Scarcity of Resources	Sense of Compelled Commitment	Sense of Futility of Effort	Sense of Negative Experience
Sense of workplace involution	1				
Sense of scarcity of resources	0.943 ***	1			
Sense of compelled commitment	0.939 ***	0.850 ***	1		
Sense of futility of effort	0.942 ***	0.848 ***	0.836 ***	1	
Sense of negative experience	0.949 ***	0.859 ***	0.855 ***	0.869 ***	1

Note: *** indicates *p* < 0.001.

**Table 6 ijerph-19-11218-t006:** Subjects’ demographic information and their *t*-tests and one-way analysis of variance.

Demographic Variables	Category	Number of People (Percentage)	Sense of Workplace Involution		Turnover Intention	
			M ± SD	*t/F*	M ± SD	*t/F*
Gender	Male	312 (54.64%)	3.61 ± 1.02	*t* (569) = −1.973, *p* < 0.05	3.57 ± 1.09	*t* (569) = −1.22, *p* = 0.22
	Female	259 (45.36%)	3.77 ± 0.83		3.67 ± 0.94	
Education level	High school or junior college and below	27 (4.73%)	4.16 ± 0.24	*F* (3, 567) = 34.752, *p* < 0.001	3.93 ± 0.57	*F* (3, 567) = 28.968, *p* < 0.001
	College	152 (26.62%)	3.98 ± 0.51		3.96 ± 0.70	
	Undergraduate	271 (47.46%)	3.78 ± 0.87		3.69 ± 0.95	
	Master and above	121 (21.19%)	2.99 ± 1.23		2.92 ± 1.17	
Nature of work unit	State-owned enterprises	42 (7.36%)	2.70 ± 0.93	*F* (7, 563) = 53.832, *p* < 0.001	2.92 ± 1.24	*F* (7, 563) = 47.713, *p* < 0.001
	Private enterprises	215 (37.65%)	4.13 ± 0.28		4.11 ± 0.43	
	Foreign-invested enterprises	53 (9.28%)	3.61 ± 0.14		4.13 ± 0.41	
	Sino-foreign joint ventures	69 (12.08%)	3.64 ± 0.16		4.09 ± 0.41	
	Public institutions	80 (14.01%)	2.48 ± 0.98		2.77 ± 1.20	
	State administrative organs	52 (9.11%)	2.46 ± 1.06		2.75 ± 1.20	
	Government	36 (6.30%)	2.51 ± 1.08		2.81 ± 1.18	
	Others	24 (4.21%)				
Area of workplace	Eastern area	349 (61.12%)	3.42 ± 0.64	*F* (3, 567) = 24.795, *p* < 0.001	3.88 ± 0.81	*F* (3, 567) = 22.552, *p* < 0.001
	Northeast area	36 (6.30%)	2.8 ± 0.986		3.19 ± 1.23	
	Central area	85 (14.89%)	2.77 ± 1.05		3.08 ± 1.23	
	Western area	101 (17.69%)	2.94 ± 0.87		3.31 ± 1.12	

**Table 7 ijerph-19-11218-t007:** Descriptive statistics of newly recruited employees’ sense of workplace involution.

Dimension	Average Value	(Statistics) Standard Deviation
Sense of scarcity of resources	3.71	1.01
Sense of compelled commitment	3.62	1.02
Sense of futility of effort	3.70	0.99
Sense of negative experience	3.70	0.97
Sense of workplace involution	3.68	0.94
Turnover intention	3.61	1.02

**Table 8 ijerph-19-11218-t008:** Correlation analysis of newly recruited employees’ sense of workplace involution and turnover intention.

	Sense of Scarcity of Resources	Sense of Compelled Commitment	Sense of Futility of Effort	Sense of Negative Experience	Sense of Workplace Involution	Turnover Intention
Sense of scarcity of resources	1					
Sense of compelled commitment	0.850 ***	1				
Sense of futility of effort	0.848 ***	0.836 ***	1			
Sense of negative experience	0.859 ***	0.855 ***	0.869 ***	1		
Sense of workplace involution	0.943 ***	0.939 ***	0.942 ***	0.949 ***		
Turnover intention	0.839 ***	0.812 ***	0.821 ***	0.849 ***	0.880 ***	1

Note: *** indicates *p* < 0.001.

**Table 9 ijerph-19-11218-t009:** Hierarchical regression results of turnover intention.

Models and Variables	Turnover Intention			
	Tier One		Tier Two	
	*β*	*t*	*β*	*t*
Gender	0.015	0.394	−0.023	−1.145
Educational level	−0.273	−7.097 ***	−0.012	−0.545
Nature of work unit	0.065	1.713	0.022	1.123
Area of workplace	−0.291	−7.613 ***	−0.043	−2.018 *
Sense of workplace involution			0.861	38.509 ***
R^2^	0.194 ***		0.778 ***	
ΔR^2^			0.584 ***	

Note: * indicates *p* < 0.05, *** indicates *p* < 0.001.

## Data Availability

Data are available upon request.

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
