# Peer review of "Development of Questionnaire on the Sense of Workplace Involution for Newly Recruited Employees and Its Relationship with Turnover Intention"

_ijerph, 2022, doi:10.3390/ijerph191811218_

Round 1
Reviewer 1 Report
Thank you dear author of the paper. I read the paper and like to appreciate you for your hard work. My observations are given below:
The abstract of the paper is unorganized and too long. It is necessary to revise the abstract accordingly.
What is the research questions of the study? What about research objectives?
There is very limited discussion of literature review that is also related to draw literature gap. The research didn't get the gap.
Where is the hypothesis of the research?
What about conceptual framework model?
There's no theoretical discuss regarding psychological or workplace issue.
The design and methodology of the study are scattered and very poor.The demographic information must be provided in a table. How did you select sample? Why you choose them? How the questionnaire was collected online or physically? There are many issues the study missed. Variable definition is missing.
The result section is comparatively well and not enough as the design is faulty.
What is the theoretical and practical contribution of the study? What about incremental contribution?
References are very poor and limited. Limitation and future directions must be taken into consideration as a whole.
Reviewer 2 Report
- The theme analysed is interesting.
- The literature review cites both older and newer studies.
- The methods used are clearly described.
- The results obtained are interesting.
I think the abstract should present only a summary of results, not the results obtained effectively for the indicators analysed.
Which is the main objective of your paper? What does your paper bring new to the literature? Please add this information in the introduction section. There is a gap in the literature that you intend to fill with your empirical investigation? Please present clearly the hypotheses of your empirical study. Presenting the variables included in the model in a table with their definition would be helpful.
Round 2
Reviewer 1 Report
Thank you dear author for the hard and the revision. You have taken a sound effort to improve it and done a lot but not to the mark.
I believe in your work and dedication but the limitation is the quality of the journal. I believe we have to maintain the quality of the journal but this paper is failed to reach that point.
